# Study on the Dynamic Relationship between Chinese Residents’ Individual Characteristics and Commercial Health Insurance Demand

**DOI:** 10.3390/ijerph20064797

**Published:** 2023-03-09

**Authors:** Ling Tian, Haisong Dong

**Affiliations:** 1School of Economics and Management, Wuhan University, Wuhan 430072, China; 2National Institute of Insurance Development, Wuhan University, Ningbo 315100, China

**Keywords:** Chinese residents’ individual characteristics, commercial health insurance demand, dynamic relationship, SVAR

## Abstract

Based on the time series data of age characteristics, household registration characteristics, gender characteristics, education characteristics, marriage characteristics and commercial health insurance density of Chinese residents from 1997 to 2020, this paper aims to explore the dynamic relationship between the individual characteristics of Chinese residents and the demand for commercial health insurance by means of impulse response and variance decomposition analysis using an SVAR model. The results show that the age characteristics, household registration characteristics, gender characteristics, education characteristics and marriage characteristics of Chinese residents have a significant impact on the demand for commercial health insurance, but there is a time lag. There is a long-term equilibrium relationship between them: In terms of age characteristics and gender characteristics, the former has a positive effect in the short term and a significant inhibition on commercial health insurance demand in the long term, while the latter has the opposite. In terms of household registration characteristics, education characteristics and marriage characteristics, there are positive effects on the whole and negative effects in a particular period.

## 1. Introduction

For a long time, it has been difficult and expensive to see a doctor in China, and the problem of poverty due to illness is still prominent. Whether or not these problems can be properly solved is directly related to the realization of China’s goal of a harmonious socialist society and the consolidation of the achievements of reform and opening up. “Medical reform” is an effective measure to solve these problems. At present, China has established four basic medical insurance systems, namely, basic medical insurance for urban workers, basic medical insurance for non-working urban residents, new rural cooperative medical insurance and social medical assistance. However, since the burden of this is mainly borne by the state and given China’s large population base and low level of economic development, it can only provide the most basic medical care for Chinese residents with a low level of security. It is unable to provide Chinese residents with higher level of protection and meet the needs of Chinese residents for personalized medical care. Therefore, it is urgent to establish and improve a multi-level medical care system to ease the financial burden of the government and provide better health care for people with different needs. To this end, commercial health insurance plays an irreplaceable role in making up for the shortage of basic medical insurance, improving the overall care level, meeting the diversified health care needs of the public, and establishing and improving a multi-level medical care system [1]. The structural framework of China’s multi-level medical care system is shown in Figure 1.

Since the commercial health insurance industry of China resumed operation in 1982, its business scale has become increasingly large and its development speed has advanced rapidly [2,3,4]. In terms of the number of companies, this increased from 1 in 1982 to 164 in 2020; in terms of premium scale, this increased from CNY 57.4 billion in 2009 to CNY 817.3 billion in 2020, and the annual compound growth rate reached 24.78%, higher than the life insurance premium income of 10.22% and accident insurance premium income of 14.55% in the same period. In terms of operating income of professional health insurance companies, take 2019 and 2020 as examples: “Ping An Health” realized a premium income of about CNY 6.534 billion and CNY 9.261 billion, respectively, increasing by 41.74%; the premium income of “PICC Health” increased from CNY 19.595 billion to CNY 27.806 billion, an increase of 41.90%; “Pacific Health” saw a year-on-year growth of 52.86%, with premium income of about CNY 2.957 billion and CNY 4.520 billion, respectively.

However, compared with the international average level, the development of China’s commercial health insurance is still lagging behind [2]. Take the proportion of commercial health insurance premium income in the life insurance market as an example: in 2020, the proportion was 24.52%, while the proportion was generally 30% in the mature insurance market. Take the proportion of commercial health insurance expenditure in the total expenditure of medical and health expenses as an example: in 2020, the total expenditure of China’s medical and health expenses was CNY 7.22 trillion, among which the commercial health insurance indemnity expenditure was only CNY 292.1 billion; this proportion is less than 4.05%, while the proportion in developed countries is generally 10%, and the proportion of the United States is 33%.Thus, it can be seen that the development of commercial health insurance in China is in the developing stage and has not reached saturation. The effective demand for China’s commercial health insurance is insufficient, and there is still much room for improvement [5].

Residents’ individual characteristics mainly include age characteristics, household registration characteristics, gender characteristics, education characteristics, marriage characteristics, income characteristics, etc. [6]. As a basic unit of society, individual residents play a very important basic role in social and economic activities, so the consumption of commercial health insurance is closely related to the individual characteristics of residents. This paper aims to empirically analyze and explore the influence degree and dynamic relationship of individual characteristics of Chinese residents on the demand for commercial health insurance by constructing an SVAR model and seeks to explain the mechanism of the influence of individual characteristics of Chinese residents on the demand for commercial health insurance from the perspective of population economics so as to provide valuable information for promoting the stable development of commercial health insurance in China. The remainder of this paper is arranged as follows: the second part is a review of the literature; the third part is theoretical analysis and research hypothesis; the fourth part is research design; the fifth part is empirical analysis; the sixth part is conclusions; the seventh part is policy enlightenments; and the last part is research prospects.

## 2. Relevant Literature Review

As for the research topic of the impact of individual characteristics of residents on the demand for commercial health insurance, scholars have conducted a large number of theoretical and empirical studies. Hopkins and Kidd (1996) [7] built the logit model and concluded through empirical analysis that age, income and health status were significantly correlated with the demand for private health insurance. Saver and Doescher (2000) [8], based on the data of the US medical expenditure survey in 1987, believed that the national characteristics, education level, disposable income and wealth level of consumers were the main factors affecting the demand for commercial health insurance. Bundorf (2002) [9] conducted an empirical study on the data of enterprise health insurance plans in 10 states and concluded that the demand for health insurance was positively related to the wage level of employees. Asgary et al. (2004) [10] took the data of 2139 households in Iran as research samples and used the conditional value assessment method (CVM) of iterative bidding game technology to draw the conclusion that the age and education level of the household head have a significant positive impact on the demand for commercial health insurance. Wang and Rosenman (2007) [11] found that education level and income have a significant impact on rural medical insurance demand. Khwaja (2010) [12] took the commercial health insurance premiums that individuals are willing to pay as the research object and adopted the dynamic random utility model for analysis. The results show that the demand for commercial health insurance was significantly positively correlated with the level of aging and negatively correlated with the level of education. Lee (2012) [13] constructed a discrete choice model and concluded that the gender, education level, marital status, household registration type and assets of residents have significant impacts on the demand for private health insurance. Oraro et al. (2018) [14] analyzed the cross-sectional data of 550 households in northwestern Cameroon and eastern Oman and concluded that male demand for commercial health insurance was highly correlated with socio-economic status, education level, age and trust in insurance companies; women’s demand for commercial health insurance was significantly related to their income and childbearing status. Wang et al. (2018) [15] collected the survey data of 1,842 households in Qinghai and Zhejiang provinces, obtained the demand for long-term care insurance (LTCI) based on the conditional valuation method, and analyzed the relevant factors of the demand for long-term care insurance (LTCI) by using logistic regression with random effects. The study found that age and education level were significantly correlated with long-term care insurance demand (LTCI). Kapur (2020) [16] collected the survey data of the Health Insurance Authority from 2009 to 2017 and empirically discussed the determinants of private health insurance demand in Ireland. It was concluded that older and sicker people were more likely to have private health insurance, that is, the age of residents was an important influencing factor on the demand for private health insurance. Using data from a unique panel of young Australian women, Doiron and Kettlewell (2020) [17] found that children have a significant negative impact on the demand for commercial health insurance, and income level also has a significant impact on the demand for commercial health insurance. Kattih and Mixon (2020) [18] built a random coefficient model and concluded that age and gender have an important impact on the demand for commercial health insurance. Xu et al. (2022) [2] built the Probit model based on the data of the China General Social Survey (CGSS) in 2017. The research results show that age, education level and income significantly promoted residents’ demand for commercial health insurance, but gender and marital status had no significant impact on the demand for commercial health insurance. Zhong et al. (2022) [19] analyzed and studied the data of the China Health and Retirement Longitudinal Study (CHARLS) in 2011, 2013 and 2015. By constructing a panel logit model, it was concluded that age and marital status of residents were significantly negatively correlated with the demand for commercial medical insurance, while gender and education level of residents were significantly positively correlated with the demand for commercial medical insurance. Call et al. (2022) [20] built the Logit model based on the comparative medical insurance measurement error survey data in the United States in 2015. Through empirical analysis, it was concluded that residents’ income level and ethnic characteristics were significantly correlated with the demand for commercial health insurance. Gao et al. (2022) [21] made an empirical study by constructing a fixed-effect panel data model and concluded that there was a significant correlation between household registration type, income and commercial health insurance demand. Jia and Yan (2022) [22] built an IV-Probit model with the cross-sectional data of the China Household Finance Survey (CHFS) in 2017 and concluded that residents’ income, age, marital status, education level and household registration type are important factors affecting the demand for commercial health insurance.

Through a review of the literature, it can be seen that although the research literature on the impact of individual characteristics of residents on the demand for commercial health insurance has been abundant enough, they are all purely “one-size-fits-all”, and the influence is “one-sided”. At present, there have been no studies in the literature that discuss the dynamic mechanism of the impact of individual characteristics of residents on the demand for commercial health insurance. Therefore, the marginal contribution of this paper is as follows: firstly, it attempts for the first time to study the influence degree of residents’ individual characteristics on commercial health insurance demand by constructing an SVAR model and excavates the dynamic relationship between residents’ individual characteristics and commercial health insurance demand; secondly, according to the research conclusions, it provides important clues for commercial health insurance companies to update their products in time, lock potential target customers, and provide relevant policies for the government to promote the harmonious and stable development of commercial health insurance in China.

## 3. Theoretical Analysis and Research Hypothesis

### 3.1. Dynamic Relationship between Residents’ Age Characteristics and Commercial Health Insurance Demand

Although the residents’ age characteristics have a significant impact on the demand for commercial health insurance, the impact of the residents’ age characteristics on the demand for commercial health insurance in China should be different in different periods. On the one hand, in the short term, although the proportion of the total dependency population is increasing, parents are willing to invest more in aspects of their children’s health security based on the current social situation influenced by long-term family planning policy and the family bearing idea, which will therefore increase the demand of commercial health insurance. Due to the declining physical function of elderly residents, their risk of disease increases. In order to mitigate this risk, the elderly will also increase their demand for commercial health insurance [12]. That is, the age characteristics of residents have a positive effect on the demand for commercial health insurance.

On the other hand, in the long term, due to the increasing proportion of the total dependent population, the elderly population or the number of minors seriously exceed the standard, resulting in excessive family economic burden and decreased purchasing power, thus reducing the demand for commercial health insurance, that is, the age characteristics of residents have a negative effect on the demand for commercial health insurance [19]. Based on the above discussion, this paper proposes the following hypothesis:

**Hypothesis 1.** 
*Residents’ age characteristics have different impacts on the demand for commercial health insurance in China in different periods, with positive effects in the short term and negative effects in the long term.*


### 3.2. Dynamic Relationship between Residents’ Household Registration Characteristics and Commercial Health Insurance Demand

Although the residents’ household registration characteristics have a significant impact on the demand for commercial health insurance, the impact of the residents’ household registration characteristics on the demand for commercial health insurance in China should be different in different periods. On the one hand, in a certain period, the proportion of the urban population increases, but among the increased urban residents, some of them are temporary residents with unstable income, or there is even an increase in the number of residents in urban slums. Due to poor living conditions, they will reduce the demand for commercial health insurance. In other words, when these residents occupy the dominant position, the characteristics of household registration have a negative effect on the demand for commercial health insurance [13].

On the other hand, on the whole, the increase in the proportion of the urban population indicates that the quality of urbanization is higher, residents have more disposable income and strong purchasing power, and after meeting basic living security, they will pursue a higher level of health security demand, so they will increase the purchase of commercial health insurance products, that is, the residents’ household registration characteristics have a positive effect on the demand for commercial health insurance [22]. Based on the above discussion, this paper proposes the following hypothesis:

**Hypothesis 2.** 
*Residents’ household registration characteristics have different impacts on the demand for commercial health insurance in China in different periods, with positive effects on the whole and negative effects in a certain period.*


### 3.3. Dynamic Relationship between Residents’ Gender Characteristics and Commercial Health Insurance Demand

Although the residents’ gender characteristics have a significant impact on the demand for commercial health insurance, the impact of the residents’ gender characteristics on the demand for commercial health insurance in China should be different in different periods. On the one hand, in the short term, although the proportion of the male population increases, men tend to be more aggressive, with low risk aversion and risk prevention awareness and weak acceptance ability of insurance. Therefore, with the increase in the male population, the consumption of commercial health insurance will decrease, that is, the gender characteristics of residents have a negative effect on the demand for commercial health insurance [5,13].

On the other hand, in the long term, men tend to be the “pillar” of the family, responsible for going out to work, earning money and supporting the family, and bear more pressure in social work and daily life, resulting in a greater degree of physical damage. For the protection of physical health, they will increase their purchase of commercial health insurance products, that is, the gender characteristics of residents have a positive effect on the demand for commercial health insurance [19]. Based on the above discussion, this paper proposes the following hypothesis:

**Hypothesis 3.** 
*Residents’ gender characteristics have different impacts on the demand for commercial health insurance in China in different periods, with negative effects in the short term and positive effects in the long term.*


### 3.4. Dynamic Relationship between Residents’ Education Characteristics and Commercial Health Insurance Demand

The residents’ education characteristics have a two-way effect on the demand for commercial health insurance. Therefore, the impact of residents’ education characteristics on the demand for commercial health insurance in China should be different in different periods. On the one hand, in a certain period, the increase in the proportion of the population with higher education means that individual residents are generally better educated and better able to understand more complex and professional commercial health insurance products. They will choose commercial health insurance products more rationally based on their own actual conditions, rather than “follow the crowd” to buy them. Therefore, the consumption of commercial health insurance products without competitive advantages will decrease, that is, when this part of residents’ thoughts occupies the dominant position, residents’ education characteristics have a negative effect on the demand for commercial health insurance [13].

On the other hand, on the whole, the increase in the proportion of the population with higher education means that they have a strong ability to accept new things actively and a high awareness of risk diversification, so they will increase the consumption of commercial health insurance. At the same time, residents with a high level of education often have relatively stable jobs and higher incomes and can afford to buy commercial health insurance products. In other words, residents’ education characteristics have a positive effect on the demand for commercial health insurance [11,22]. Based on the above discussion, this paper proposes the following hypothesis:

**Hypothesis 4.** 
*Residents’ education characteristics have different impacts on the demand for commercial health insurance in China in different periods, with positive effects on the whole and negative effects in a certain period.*


### 3.5. Dynamic Relationship between Residents’ Marriage Characteristics and Commercial Health Insurance Demand

Residents’ marriage characteristics have a two-way effect on the demand for commercial health insurance. Therefore, the impact of marriage characteristics on the demand for commercial health insurance in China should be different in different periods. On the one hand, in a certain period, the proportion of the married population increases, which means that the number of married residents will increase, so it is necessary to consider the problems of supporting parents and children, and the economic burden is large. Some residents will reduce the consumption of commercial health insurance, that is, when this idea of the residents dominates, residents’ marriage characteristics will have a negative effect on the demand for commercial health insurance [19].

On the other hand, on the whole, the increase in the number of married residents, due to the increase in economic pressure, will increase their attention to the health of family members, and their awareness of risk will be stronger, within the scope of ability to increase the purchase of commercial health insurance products, that is, the residents’ marriage characteristics have a positive effect on the demand for commercial health insurance [5,22]. Based on the above discussion, this paper proposes the following hypothesis:

**Hypothesis 5.** 
*Residents’ marriage characteristics have different impacts on the demand for commercial health insurance in China in different periods, with positive effects on the whole and negative effects in a certain period.*


## 4. Study Design

### 4.1. Variable Specifications

With regard to commercial health insurance demand, premiums, amount insured, value of policy in force, number of insured, density of insurance and depth of insurance are usually the main indicators of insurance demand in a country or region. Because of the availability and comparability of data in this paper, insurance density was chosen to measure insurance demand. This paper uses the density of commercial health insurance as a quantitative indicator of the demand for commercial health insurance.

Residents’ individual characteristics: the proportion of the total dependent population was selected as the quantitative index of residents’ age characteristics; the proportion of urban population was selected as the quantitative index of residents’ household registration characteristics; the proportion of the male population was selected as the quantitative index of residents’ gender characteristics; the proportion of the population post-higher education was selected as the quantitative index of residents’ education characteristics; the proportion of the married population was selected as the quantitative index of residents’ marriage characteristics.

The name, unit of measurement, symbol and definition of each variable are shown in Table 1:

### 4.2. Data Introduction

This paper uses the annual time series data from 1997 to 2020 at the national level to study the dynamic relationship between the commercial health insurance demand of Chinese residents and the five variables selected above that represent residents’ individual characteristics. “healthins” data are from the official website of the China Banking and Insurance Regulatory Authority (CBIRC) [23]. Data on “age”, “urban”, “sex”, “edu” and “married” are all from the China Statistical Yearbook [24]. The data preprocessing software was Excel and the econometric analysis software was Stata. The final sample size of this paper was 24. In order to minimize the impact of heteroscedasticity on the empirical model, natural logarithms were adopted for all variables with currency as the unit of measurement. In order to eliminate the influence of price changes, all nominal variable data expressed in currency were adjusted by using 1997 as the base period.

### 4.3. Model Setting

#### 4.3.1. Reasons for Using the SVAR Model

The vector autoregressive model (VAR) adopts the simultaneous method of multiple equations. In each equation of the model, a single endogenous variable carries out regression on the lag term of all endogenous variables in the model so as to estimate the dynamic relationship of all endogenous variables in the model. However, the VAR model does not set any constraint conditions and simply reflects the dynamic relationship between endogenous variables. It does not explicitly explain the meaning of economic structure among endogenous variables in the model, that is, it ignores the implied economic structure among endogenous variables and the influence relationship of endogenous variables over the same period to a certain extent [25]. Based on these defects of the VAR model, in this paper we constructed a 6-variable SVAR(P) model to analyze the dynamic influence mechanism of residents’ age characteristics, residents’ household registration characteristics, residents’ gender characteristics, residents’ education characteristics and residents’ marriage characteristics on the demand for commercial health insurance.

#### 4.3.2. Introduction to SVAR Model

The SVAR model can be derived from the simplified VAR model. The model expression of M-variable VAR(P) is:(1)$$Y_{t}=\phi_{1}Y_{t-1}+\phi_{2}Y_{t-2}+\cdots +\phi_{p}Y_{t-p}+u_{t},$$
where *Y_t_* is the vector “M × 1” and *u_t_* is the vector of random perturbation term in simplified expression, allowing the existence of concurrent correlation.

Multiply both sides of “Equation (1)” by a non-degenerate matrix “*A*” in the left at the same time, and transfigure the terms to obtain:(2)$$A(I-\phi_{1}L-\phi_{2}L^{2}-\cdots -\phi_{p}L^{p})Y_{t}=Au_{t},$$

Suppose *B* is an “M × M” matrix; then, “Equation (2)” can be written as:(3)$$A(I-\phi_{1}L-\phi_{2}L^{2}-\cdots -\phi_{p}L^{p})Y_{t}=Au_{t}=B\varepsilon_{t},$$
where *ε_t_* is an orthonormal random perturbation term vector, that is, the covariance matrix of *ε_t_* is normalized to the identity matrix *I*_M_.

“Equation (3)” is called “AB model” of SVAR.

#### 4.3.3. SVAR Model Expression

Since the optimal lag period of the model was verified as 2 in the following part, this paper constructs a 6-variable SVAR(2) model, whose model expression is as follows:$$AY_{t}=\delta +\phi_{1}Y_{t-1}+\phi_{2}Y_{t-2}+\varepsilon_{t},$$
$$Where,\;Y_{t}=\left[\begin{array}{c} {\mathrm{age}}_{t}\\ {\mathrm{urban}}_{t}\\ {\mathrm{sex}}_{t}\\ edu_{t}\\ married_{t}\\ \ln healthins_{t} \end{array}\right];\;A=\left[{}_{}\begin{array}{cccccc} 1 & -c_{12} & -c_{13} & -c_{14} & -c_{15} & -c_{16}\\ -c_{21} & 1 & -c_{23} & -c_{24} & -c_{25} & -c_{26}\\ -c_{31} & -c_{32} & 1 & -c_{34} & -c_{35} & -c_{36}\\ -c_{41} & -c_{42} & -c_{43} & 1 & -c_{45} & -c_{46}\\ -c_{51} & -c_{52} & -c_{53} & -c_{54} & 1 & -c_{56}\\ -c_{61} & -c_{62} & -c_{63} & -c_{64} & -c_{65} & 1 \end{array}\right];$$
$$\delta =\left[\begin{array}{c} c_{1}\\ c_{2}\\ c_{3}\\ c_{4}\\ c_{5}\\ c_{6} \end{array}\right];\;\phi_{i}=\left[\begin{array}{cccccc} \gamma_{11}^{\left(i\right)} & \gamma_{12}^{\left(i\right)} & \gamma_{13}^{\left(i\right)} & \gamma_{14}^{\left(i\right)} & \gamma_{15}^{\left(i\right)} & \gamma_{16}^{\left(i\right)}\\ \gamma_{21}^{\left(i\right)} & \gamma_{22}^{\left(i\right)} & \cdots & \cdots & \cdots & \gamma_{26}^{\left(i\right)}\\ \gamma_{31}^{\left(i\right)} & \vdots & \ddots & & & \gamma_{36}^{\left(i\right)}\\ \gamma_{41}^{\left(i\right)} & \vdots & & \ddots & & \gamma_{46}^{\left(i\right)}\\ \gamma_{51}^{\left(i\right)} & \vdots & & & \ddots & \gamma_{56}^{\left(i\right)}\\ \gamma_{61}^{\left(i\right)} & \gamma_{62}^{\left(i\right)} & \gamma_{63}^{\left(i\right)} & \gamma_{64}^{\left(i\right)} & \gamma_{65}^{\left(i\right)} & \gamma_{66}^{\left(i\right)} \end{array}\right],\;i=1,\;2;\;\varepsilon_{t}=\left[\begin{array}{c} \varepsilon_{1t}\\ \varepsilon_{2t}\\ \varepsilon_{3t}\\ \varepsilon_{4t}\\ \varepsilon_{5t}\\ \varepsilon_{6t} \end{array}\right]$$

## 5. Empirical Analysis

### 5.1. Stationarity Test and Cointegration Test of Variable Data

Some unstable time series often show a common trend of change, but these time series are not actually related, resulting in a “pseudo-regression” problem. Based on the time series data adopted in this paper, it was necessary to conduct a stationarity test for each time series variable. The stability test method of variable data used in this paper is the ADF unit root test [26,27,28,29,30,31], and the test results of each variable are shown in Table 2. It can be seen from Table 2 that none of the variables passed the stationarity test, that is, the original time series is unstable. After first-order difference, the P value of the ADF test for first-order difference variable “D.age” is 0.0232, which passes the significance level test of 5%. The ADF test of first-order difference variable “D.urban” has a P value of 0.0632, which is significant at the significance level of 10%. The P values of the ADF test for the remaining first-order difference variables “D.sex”, “D.edu”, “D.maried” and “D.healthins” are 0.0052, 0.0000, 0.0004 and 0.0001, respectively, which pass the 1% significance level test, indicating that all variables are first-order unitary. The necessary condition for the existence of long-term equilibrium relationship is satisfied.

Subsequently, in order to further verify whether there is a long-term equilibrium relationship between the data of various variables, the Johansen co-integration relationship test [27,30,32] was carried out, and the test results are shown in Table 3. According to the trace test statistics value in Table 3, there are two linearly independent co-integration vectors (asterisk “*” in the table), which accept the null hypothesis that the co-integration rank is 2, that is, two co-integration relationships exist in these six time series variables. These six variables do have a long-term equilibrium relationship, and their linear combination is stable, thus ensuring the accuracy of subsequent empirical analysis.

### 5.2. Determination of the Optimal Lag Period of the Model

On the one hand, the larger the number of lag periods is, the better it can reflect the dynamic characteristics of the model; on the other hand, the larger the number of lag periods is, the more parameters need to be estimated, which makes the effective sample size too small, the estimation error large, and the prediction accuracy lower. Therefore, it is very necessary to determine the optimal number of lag periods of the SVAR model when both are taken into account. In this paper, four information criteria (FPE, AIC, HQIC and SBIC) were used to select the optimal number of lag periods of the SVAR model [26,27,28,31,33]. According to the evaluation criterion that the smaller the value of an information criterion is, the better the result will be, the optimal lag period was finally determined to be 2 (asterisk “*” in the table), as shown in Table 4. Therefore, the six-variable SVAR(2) model was established in this paper.

### 5.3. Define Constraint Matrix

In order to identify the “AB model” of SVAR, constraints need to be imposed on the matrices “A” and “B”. Chen (2014) [34] and Cheng et al. (2022) [26] pointed out that for the “AB model” with M endogenous variables, [2M^2^ − M(M + 1)/2] constraints need to be imposed on matrix “A” and “B” to identify the AB model. Therefore, 51 constraints need to be set in this paper (See Table A4 in Appendix A of this paper for the parameter estimation results of model).

In this paper, short-term constraints were applied, that is, constraints are applied to matrix “A” and “B”. The constraint method is to follow the idea of Cholesky decomposition: Matrix “A” is set as the lower triangular matrix and all the main diagonal elements are 1. Matrix “B” is set as a diagonal matrix [26,34,35]. So the constraint matrix “A” and “B” are defined as follows:$$A=[\begin{array}{cccccc} 1 & 0 & 0 & 0 & 0 & 0\\ . & 1 & 0 & 0 & 0 & 0\\ . & . & 1 & 0 & 0 & 0\\ . & . & . & 1 & 0 & 0\\ . & . & . & . & 1 & 0\\ . & . & . & . & . & 1 \end{array}];\;B=\left[\begin{array}{cccccc} . & 0 & 0 & 0 & 0 & 0\\ 0 & . & 0 & 0 & 0 & 0\\ 0 & 0 & . & 0 & 0 & 0\\ 0 & 0 & 0 & . & 0 & 0\\ 0 & 0 & 0 & 0 & . & 0\\ 0 & 0 & 0 & 0 & 0 & . \end{array}\right]$$

### 5.4. Model Stability Test

The sufficient and necessary condition for the stability of the SVAR model is that all the characteristic roots of the characteristic equation are in the unit circle [26,34,36], that is, the reciprocal of the roots of the polynomial equation with autoregressive coefficient are in the unit circle. In this part of the paper, model stability is tested (see Figure 2 for details). Since the model contains six endogenous variables and the optimal lag period is 2, the model has 12 eigenvalues. Figure 2 shows that all the 12 eigenvalues are within the unit circle, which indicates that the six-variable SVAR(2) model constructed in this paper is stable and the sample size is sufficient, which ensures the effectiveness of subsequent impulse response and variance decomposition analysis in this paper.

### 5.5. Impulse Response Analysis

First of all, since the SVAR model was adopted in this paper (the verifications of the basic assumptions of the SVAR model are shown in Table A2 and Table A3 in Appendix A of this paper), which reflects the economic structure relationship between variables [25], the Granger causality test was not necessary. Based on the research purpose and empirical research, this paper recognizes the following conclusions: “age”, “sex”, “married”, “urban” and “edu” are the Granger causes of “lnhealthins”.

We followed with the impulse response analysis. Impulse response function analysis accurately describes the dynamic influence path of each endogenous variable on itself and other endogenous variables when it is impacted. Figure 3, Figure 4, Figure 5, Figure 6 and Figure 7 shows the impulse response function diagram of variable “lnhealthins” impacts by various factor variables of residents’ individual characteristics. In Figure 3, Figure 4, Figure 5, Figure 6 and Figure 7, the abscissa represents the number of tracking periods of the impulse response function, and the ordinate represents the response degree of response variable to impact variable. The following is a specific analysis:

Figure 3 shows the impulse response function diagram of “lnhealthins” under the impact of “age”. After giving “age” a positive impact in the current period, “lnhea- lthins” has a positive response of about 0.75 in the first period and reaches the highest point in this period, shows negative changes in the second period, and reaches the lowest point of −0.25 in the fifth period. From then on, there is a small negative fluctuation until the 9th stage. From the tenth stage and later, the impact effect basically disappears and stays near 0. In other words, in the short term, “age” has a significant positive effect on “lnhealthins”; in the long term, “age” has a significant negative effect on “lnhealthins”. Thus, Hypothesis 1 can be verified.

Figure 4 shows the pulse response function diagram of “lnhealthins” under the impact of “urban”. After a positive impact on urban in the current period, “lnhealthins” changes slightly and alternately in the first seven periods, reaching the maximum positive effect of 0.05 in the 3rd period, and showing a negative effect from the 7th to the 12th period. It begins to grow slowly and positively after the 12th period. In other words, in general, “urban” has a significant positive effect on “lnhealthins”, although it may have a negative effect at some time. Thus, Hypothesis 2 can be verified.

Figure 5 shows the pulse response function diagram of “lnhealthins” under the impact of “sex”. After a positive impact on sex in the current period, the negative and positive effects of lnhealthins in the first seven periods alternates successively, but the negative effect lasts for a long time, and reaches the lowest point of negative effect −0.03 in the first period. The positive effect peaks at 0.02 in the third phase. After the seventh phase, the positive effect begins to increase rapidly again, and its value basically maintains the peak of the positive effect in the third phase. In other words, in the short term, “sex” has a significant negative effect on “lnhealthins”; in the long term, “sex” has a significant positive effect on “lnhealthins”. Thus, Hypothesis 3 can be verified.

Figure 6 shows the pulse response function diagram of “lnhealthins” under the impact of “edu”. After a positive impact on “edu” in the current period, “lnhealthins” continues to grow positively, reaching the maximum value of 0.04 in the 4th period, and then begins to fall back, falling to 0 in the 7th period, and growing negatively from this period to the 14th period. In the 10th period, the negative effect decreases to −0.02, and the growth trend begins to be positive after the 14th period. In other words, in general, “edu” has a significant positive effect on “lnhealthins”, although it may have a negative effect at some time. Thus, Hypothesis 4 can be verified.

Figure 7 shows the impulse response function diagram of “lnhealthins” under the impact of “married”. After a positive impact on “married” in the current period, the first four phases of “lnhealthins” shows a positive correlation, the positive impact effect reaches the maximum (0.03) in the third phase, and the negative impact effect from the fourth to the seventh phase. However, the negative effect value is relatively small and stays near 0 in the later stage, and presents a positive growth trend after the seventh stage. In other words, in general, “married” has a significant positive effect on “lnhealthins”, although it may present a negative effect in a certain period of time. Thus, Hypothesis 5 can be verified.

### 5.6. Variance Decomposition Analysis

Variance decomposition analysis quantitatively and roughly describes the influence relationship between endogenous variables and further evaluates the importance of each endogenous variable by analyzing the contribution of structural impact of each endogenous variable to the change of all endogenous variables (usually measured by variance). Columns (1) to (5) in Table 5 successively show the variance decomposition results of variables “lnhealthins” impacted by the residents’ individual characteristics representing variables “age”, “urban”, “sex”, “edu” and “married”.

It can be seen from Table 5 that the variables represented by residents’ individual characteristics, “age”, “urban”, “sex”, “edu” and “married”, all have a short time lag on the impact of the endogenous variables of “lnhealthins”, and “lnhealthins” is not affected by them in the current and the first period. With the passage of time, the relative variance contribution of “age”, “urban”, “sex”, “edu” and “married” to “lnhealthins” increases gradually. However, the relative variance contribution rates of these five representative variables to “lnhealthins” are different. Compared with the maximum relative variance contribution rates, from the largest to the smallest, they are “age” (the 8th phase, 23.98%), “urban” (the 6th phase, 18.56%), “edu” (the 13th phase, 11.74%), “married” (the 5th phase, 6.72%) and “sex” (the 15th phase, 5.57%), that is, the importance of these five endogenous variables on “lnhealthins” from low to high is “sex”, “married”, “edu”, “urban” and “age”.

## 6. Conclusions

Based on the time series data of 24 years from 1997 to 2020, in this paper we selected five variables representing individual characteristics of residents, namely, age characteristics, household registration characteristics, gender characteristics, education characteristics and marriage characteristics, and constructed an SVAR model in order to explore the dynamic relationship between the individual characteristics of Chinese residents and the demand for commercial health insurance. The following six research conclusions are obtained:(1)The five representative variables of residents’ individual characteristics all have an impact on the demand for commercial health insurance, but the degree of importance is different. The importance, from high to low, is residents’ age characteristics, household registration characteristics, education characteristics, marriage characteristics and sex characteristics;(2)The age characteristics, household registration characteristics, education characteristics, marriage characteristics and gender characteristics of residents have time lag on the demand for commercial health insurance;(3)There is a long-term equilibrium relationship between the six variables, that is, some linear combination of the six variables is stable;(4)In terms of age characteristics of residents, there is a positive effect in the short term, and significantly inhibits the demand for commercial health insurance in the long term;(5)In terms of household registration characteristics, education characteristics and marriage characteristics, there is a positive effect on the whole, but a negative effect in a certain period;(6)In terms of gender characteristics of residents, there is a negative effect in the short term and a significant promotion of the demand for commercial health insurance in the long term.

## 7. Policy Enlightenments

Based on the above main research conclusions, the policy enlightenments of this paper are as follows:(1)Strengthen the research and development of commercial health insurance products to meet the health security needs of residents of different ages. The specific measures are as follows: Firstly, in terms of elderly residents, insurance companies should develop commercial health insurance products according to the physiological and psychological characteristics of the elderly residents, such as long-term care insurance, silver health insurance and serious disease insurance. Secondly, in terms of children and adolescents, insurance companies should actively promote children’s health insurance, harmonious and healthy children’s insurance, infant health insurance products, etc., in view of the fact that minors are the “heart of the family” and have their own unique characteristics;(2)Strive to improve the education level of residents and optimize the educational structure of the population. The specific measures are as follows: Firstly, strengthen the construction of rural basic education facilities, introduce high-level talents into rural areas, promote educational exchanges between rural and urban areas, and narrow the gap between urban and rural education resources. Secondly, increase the investment in educational resources, deepen the reform of the education system, increase the number of colleges and universities, and build a high-quality higher education personnel training system;(3)Actively promote the transformation of the rural population into urban residents and effectively improve the consumption level of residents. The specific measures are as follows: Firstly, speed up the urbanization process of the household registration population; improve the medical, serious disease, nursing disability and other health security systems; promote the development of urbanization; and improve the quality of urbanization. Secondly, promote employment by attracting investment, actively develop tertiary industry, increase residents’ disposable income, and optimize residents’ consumption structure;(4)Encourage early marriage and childbearing and advocate a happy and harmonious family. The specific measures are as follows: Firstly, in view of the current situation of aging and fewer children in China, improve relevant laws and regulations, actively publicize and implement the “open three-child” policy, guide young people of appropriate age to get married as early as possible, and encourage newlywed families to have more children so as to achieve a long-term balance of population age structure. Secondly, actively promote the construction of a harmonious family featuring “love between husband and wife, respect for the old and care for the young, and harmony among neighbors”, strive to improve the quality of life of women, and rely on women’s progress in civilization to drive the harmony of the family.

## 8. Research Prospects

When studying the impact of residents’ individual characteristics on the demand for commercial health insurance, this paper selects five representative variables, namely the characteristics of residents’ household registration, residents’ age, residents’ gender, residents’ education and residents’ marriage characteristics, and investigates the dynamic transmission mechanism of their impact on the demand for commercial health insurance. However, individual characteristics of residents are not limited to this, but also include psychological characteristics, ethnic characteristics, religious characteristics, personality characteristics and income characteristics of residents, etc. Due to the feasibility of the empirical model selected and the acquisition of data matching, this paper does not discuss these factors in these aspects, and these factors are rarely covered in the existing literature. Therefore, this can be regarded as the future research direction of scholars at home and abroad. The authors hold that the data of resident personality characteristics and other data representing individual characteristics of residents can be obtained by using the Myers-Briggs Type Indicator (MBTI) classification scale and the design of relevant questionnaires, and, accordingly, appropriate empirical methods and models can be selected for further research.

## Figures and Tables

**Figure 1 ijerph-20-04797-f001:**
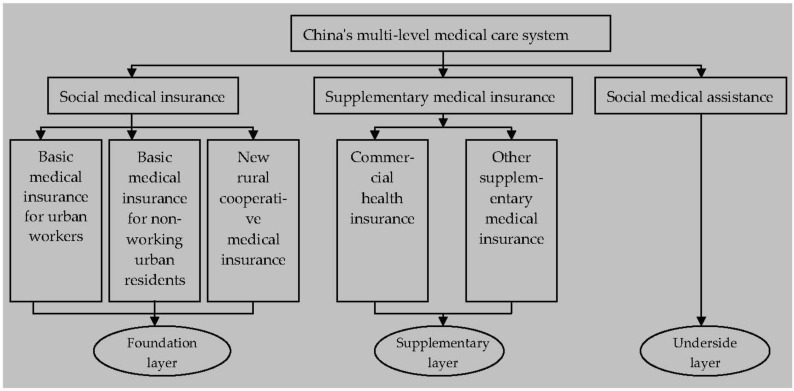
Framework of China’s multi-tiered medical care system.

**Figure 2 ijerph-20-04797-f002:**
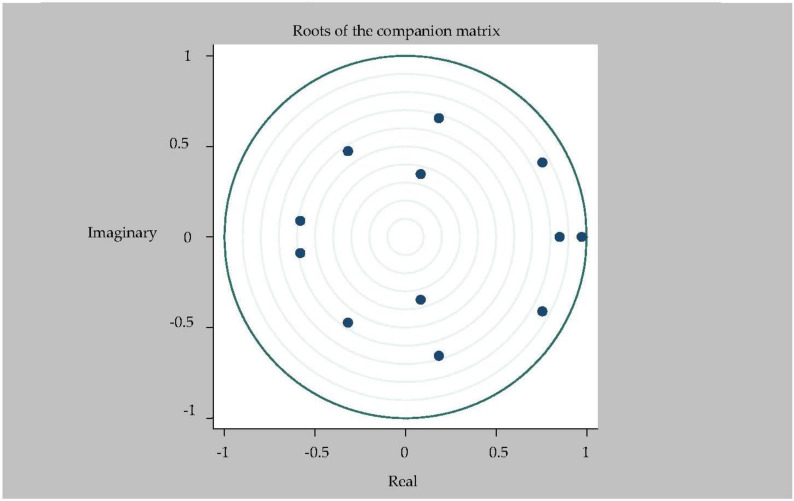
Root mode distribution of characteristic equation of 6-variable SVAR(2) model.

**Figure 3 ijerph-20-04797-f003:**
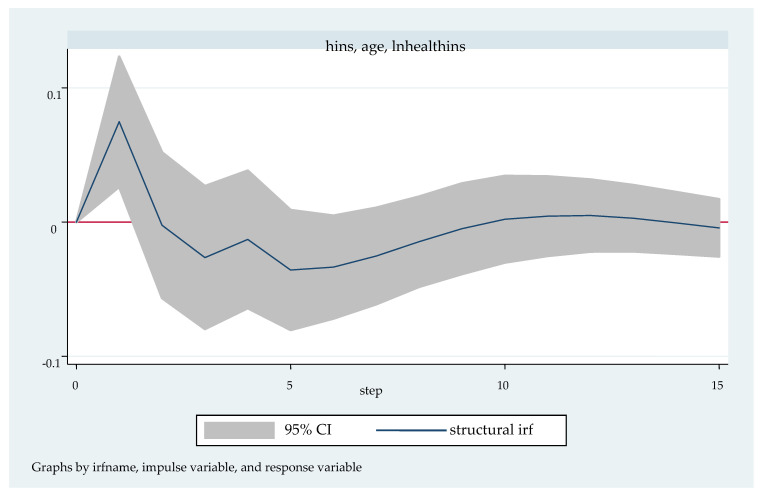
Impulse response function diagram of “lnhealthins” impacted by “age”.

**Figure 4 ijerph-20-04797-f004:**
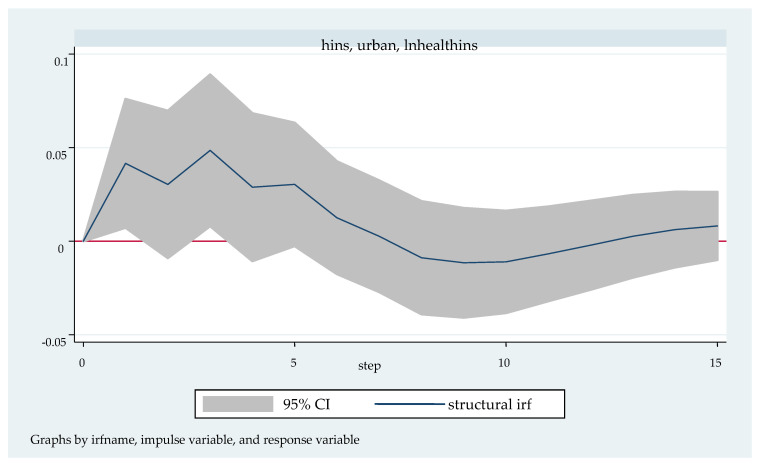
Impulse response function diagram of “lnhealthins” impacted by “urban”.

**Figure 5 ijerph-20-04797-f005:**
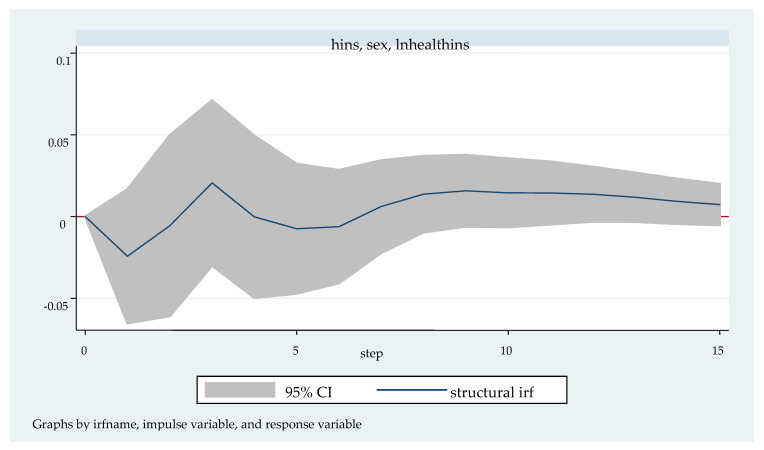
Impulse response function diagram of “lnhealthins” impacted by “sex”.

**Figure 6 ijerph-20-04797-f006:**
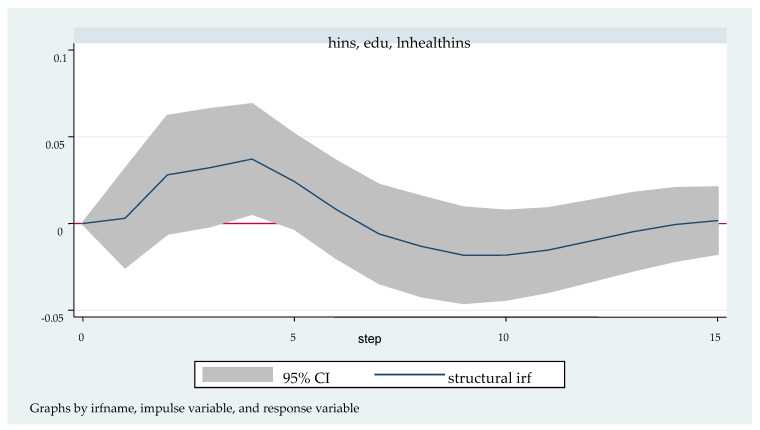
Impulse response function diagram of “lnhealthins” impacted by “edu”.

**Figure 7 ijerph-20-04797-f007:**
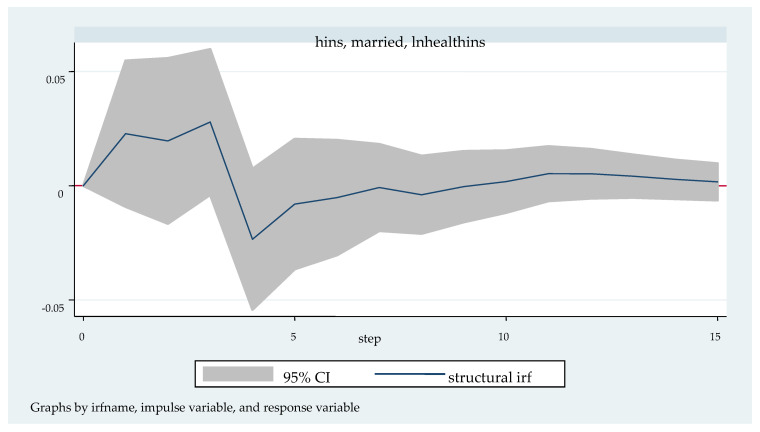
Impulse response function diagram of “lnhealthins” impacted by “married”.

**Table 1 ijerph-20-04797-t001:** Definition of each variable.

Name of Variable	Symbol of Variable	Explanation of Variables
Health insurance demand (CNY/person)	healthins	Health insurance density, namely, per capita health insurance premium income of residents
Age characteristics (%)	age	Total dependency population ratio, namely, the proportion of people aged 0–14 and over 64 in the total population of the country aged 15–64
Household registration characteristics (%)	urban	Urban population ratio, namely, the proportion of urban population in the total population of the country
Gender characteristics (%)	sex	Male population ratio, namely, the proportion of men in the total population of the country
Education characteristics (%)	edu	Higher education population ratio, namely, the proportion of the population with associate degree or above in the total population of the country aged 6 and above
Marriage characteristics (%)	married	Married population ratio, namely, the proportion of married people in the total population of the country aged 15 and above

**Table 2 ijerph-20-04797-t002:** Stability test results of each variable datum.

Variable	ADF Value	10% Critical Value	*p* Value	Result
age	−2.213	−2.630	0.2015	Unstable
urban	0.430	−2.630	0.9826	Unstable
sex	−1.891	−2.630	0.3365	Unstable
edu	0.671	−2.630	0.9892	Unstable
married	−1.252	−2.630	0.6508	Unstable
lnhealthins	−1.698	−2.630	0.4319	Unstable
D.age	−3.147	−2.630	0.0232 **	Stable
D.urban	−2.767	−2.630	0.0632 *	Stable
D.sex	−3.633	−2.630	0.0052 ***	Stable
D.edu	−5.668	−2.630	0.0000 ***	Stable
D.married	−4.300	−2.630	0.0004 ***	Stable
D.lnhealthins	−4.765	−2.630	0.0001 ***	Stable

Note: ***, ** and * are significant at the level of 1%, 5% and 10% respectively.

**Table 3 ijerph-20-04797-t003:** Johansen co-integration relationship test of variables.

H_0_: Number of Co-Integrated Ranks	Eigenvalue	Statistic Value of Trace Test	5% Critical Value	Result
0		131.3985	94.15	Reject H_0_
1	0.92878	73.2736	68.52	Reject H_0_
2	0.74086	43.5648 *	47.21	Accept H_0_

**Table 4 ijerph-20-04797-t004:** SVAR model’s optimal lag period decision test.

Number of Lag Periods	FPE	AIC	HQIC	SBIC
0	0.000106	7.87148	7.92196	8.16973
1	1.7 × 10^−9^	−3.38247	−3.02915	−1.29477
2	2.8 × 10^−10^ *	−6.8186	−6.16243	−2.94143
3		−315.126	−314.167	−309.46
4		−329.092	−328.133	−323.426

**Table 5 ijerph-20-04797-t005:** Variance decomposition results of variable lnhealthins impacted by various factor variables.

Step	(1)	(2)	(3)	(4)	(5)
	Age	Urban	Sex	Edu	Married
0	0	0	0	0	0
1	0	0	0	0	0
2	0.278797	0.086143	0.029317	0.000456	0.025807
3	0.227932	0.107755	0.025223	0.032355	0.03672
4	0.208833	0.165843	0.034603	0.060683	0.055686
5	0.194875	0.175836	0.031463	0.096849	0.067212
6	0.212659	0.185613	0.030199	0.104409	0.063053
7	0.231472	0.180492	0.029677	0.100874	0.060635
8	0.239762	0.174603	0.029635	0.098405	0.058615
9	0.239132	0.172279	0.033544	0.100278	0.057568
10	0.235033	0.172138	0.038908	0.10642	0.056445
11	0.231355	0.172286	0.043307	0.112612	0.055611
12	0.228417	0.170845	0.047553	0.116473	0.055447
13	0.226213	0.168889	0.051307	0.117422	0.055408
14	0.224372	0.167529	0.054052	0.116887	0.055311
15	0.222913	0.167302	0.055671	0.116131	0.055125

## Data Availability

The raw data are included in Appendix A of this paper (see Table A1).

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
