# Peer review of "Study on the Dynamic Relationship between Chinese Residents’ Individual Characteristics and Commercial Health Insurance Demand"

_ijerph, 2023, doi:10.3390/ijerph20064797_

Round 1

Reviewer 1 Report

This paper investigates the relationship between individual characteristics and private health care insurance demand in China. The paper provides some information regarding the Chinese situation but should be improved.

1. For the international readers, a more thorough background regarding the Chinese health care system and in particular how the public health care system influences the demand for private health care insurance.

2. More focus on methods in the literature review. The present study is a time-series analysis, but many of the cited studies are not. Individual level data provides more opportunities for explanation, and this has to be addressed in the present paper. 

3. Greater efforts should be made to relate the hypotheses of the paper to the existing literature.

Author Response

Dear reviewer:

Thank you very much for your advice. Please refer to the attachment for specific reply.

Reviewer 2 Report

Dear authors,

Thank you very much for this informative and very interesting article. I enjoyed reading it very much, also because the article is well written and understandable to read. I also find the topic highly interesting and relevant to research. Overall, I have only small suggestions that will hopefully help to improve the already very successful research presentation.

The article begins with a stringent introduction. The objective is formulated clearly and comprehensibly. The literature review in section 2 is also convincing. The state of research is well elaborated on the basis of relevant and current studies. The hypothesis derivation is also plausible. The arguments appear coherent. However, it would be advisable at this point to cite relevant literature sources to provide scientific support for the reasoning.

Methodology and statistical analysis are well described. I have no critical comments on this. I also consider the data basis for the analysis to be well chosen, so I have no concerns in this respect either. Nevertheless, it strikes me that the sample used for the analysis is quite small. This should be taken up in a chapter on limitations. I was surprised by the lack of a corresponding section anyway.

The final chapter is satisfactory. Nevertheless, not all conclusions can be derived from the results of the statistical analysis. I therefore recommend to formulate the implications more carefully by clearly indicating which statements are not supported by the analysis.

Thank you again for this very well-done contribution. I wish you success for the further revision.

Author Response

Dear reviewer,

Thank you very much for your advice. Please refer to the attachment for specific reply.

Reviewer 3 Report

This paper applies SVAR and impulse response analysis to explore the dynamic relationship between population statistics and the demand for commercial health insurance. The research topic is interesting, but there are some major concerns that should be addressed before the recommendation for publication can be made.

1. Authors should provide the results of statistical tests to verify the model's assumption. For example, tests on residuals to verify if they are uncorrelated (or orthonormal random perturbations as claimed on lines 318-319 on page 8).

2. Authors should provide all the parameter identification results in the model (the parameters on lines 325-326 on page 8), and comment on the results. Especially the interpretation of the phi matrices in line 326 should be included.

3. It looks like there are many unknown parameters that need identification. Based on the short panel dataset, the authors should evaluate if the parameter estimation results are reliable. A bootstrap method can be applied to create some confidence intervals when impulse response analysis is performed. Based on a single curve in each of Figure 2, it's not persuasive enough to make conclusions based on the small dataset.

4. Authors may carefully revise the policy enlightenments section. It looks like some are overclaimed based on the findings in this paper. 

Author Response

Dear reviewer,

Thank you very much for your advice.Please refer to the attachment for specific reply.

Round 2

Reviewer 1 Report

The authors have answered most of the concerns I have regarding the paper. However, it is not easy to draw very firm conclusions using a data with so few data points. 

Reviewer 3 Report

Thanks for revising the manuscript, but I still have a few major concerns.

1. Authors should provide the results of statistical tests to verify the model's assumption. For example, tests on residuals to verify if they are uncorrelated. This is the "assumption", not the "definition". Tests for autocorrelation and normality for residuals are very common in time series. var, svar model should not be exceptional. For example, the LM test proposed in Johansen, S. 1995. Likelihood-Based Inference in Cointegrated Vector Autoregressive Models. Oxford: Oxford University Press.

2. The confidence intervals are still necessary for impulse response analysis. Authors can use dashed lines to illustrate the upper and lower bounds, instead of using shades. Descriptions and comments about the bootstrapping results should be included in the manuscript as well.

3. Authors have to smooth the context and clearly define the terminologies used in the manuscript. According to WHO, "health security is defined as the activities required, both proactive and reactive, to minimize the danger and impact of acute public health events that endanger people’s health across geographical regions and international boundaries." It looks that the health security mentioned in the manuscript means a different thing. Also, does the phrase "medical security system" mean "health insurance system"?

4. Authors' answer to the sample size is fine, but it is necessary to include comments on the sufficiency of sample size in the manuscript.

Round 3

Reviewer 3 Report

In Table A2 and Table A3, change "accept H0" to "Fail to reject H0".
